# Restricted Activation of the NF-κB Pathway in Individuals with Latent Tuberculosis Infection after HIF-1α Blockade

**DOI:** 10.3390/biomedicines10040817

**Published:** 2022-03-31

**Authors:** Aline de Oliveira Rezende, Rafaella Santos Sabóia, Adeliane Castro da Costa, Diana Messala Pinheiro da Silva Monteiro, Adrielle Zagmignan, Luis Ângelo Macedo Santiago, Rafael Cardoso Carvalho, Paulo Vitor Soeiro Pereira, Ana Paula Junqueira-Kipnis, Eduardo Martins de Sousa

**Affiliations:** 1Graduate Program in Health Sciences, Federal University of Maranhão—UFMA, São Luís 65080-805, Brazil; aline.rezende@discente.ufma.br (A.d.O.R.); carvalho.rafael@ufma.br (R.C.C.); paulo.soeiro@ufma.br (P.V.S.P.); 2Graduate Program in Microbial Biology, CEUMA University—UniCEUMA, São Luís 65075-120, Brazil; rafaella021480@ceuma.com.br; 3Faculty Estácio de Sá de Goiás, Goiânia 74055-190, Brazil; adeliane.costa@estacio.br; 4Graduate Program in Health and Services Management, CEUMA University—UniCEUMA, São Luís 65075-120, Brazil; diana022247@ceuma.com.br (D.M.P.d.S.M.); adrielle.zagmignan@ceuma.br (A.Z.); 5Graduate Program in Biodiversity and Biotechnology, Amazônia-BIONORTE, Federal University of Maranhão—UFMA, São Luís 65080-805, Brazil; luis.angelo@ufma.br; 6Institute of Tropical Pathology and Public Health, Federal University of Goiás, Goiânia 74605-050, Brazil; ana_kipnis@ufg.br

**Keywords:** tuberculosis, latent tuberculosis infection, hypoxia-inducible factor 1 alpha (HIF-1α), tumor necrosis factor-alpha (TNF-α), nuclear factor kappa B (NF-κB)

## Abstract

Tuberculous granuloma formation is mediated by hypoxia-inducible factor 1 alpha (HIF-1α), and is essential for establishing latent tuberculosis infection (LTBI) and its progression to active tuberculosis (TB). Here, we investigated whether HIF-1α expression and adjacent mechanisms were associated with latent or active TB infection. Patients with active TB, individuals with LTBI, and healthy controls were recruited, and the expression of cytokine genes *IL15*, *IL18*, *TNFA*, *IL6*, *HIF1A*, and *A20* in peripheral blood mononuclear cells (PBMCs) and serum vitamin D (25(OH)D3) levels were evaluated. Additionally, nuclear factor kappa B (NF-κB) and tumor necrosis factor-alpha (TNF-α) levels were analyzed in PBMC lysates and culture supernatants, respectively, after HIF-1α blockade with 2-methoxyestradiol. We observed that IL-15 expression was higher in individuals with LTBI than in patients with active TB, while IL-18 and TNF-α expression was similar between LTBI and TB groups. Additionally, serum 25(OH)D3 levels and expression of *IL-6*, *HIF1A*, and *A20* were higher in patients with active TB than in individuals with LTBI. Moreover, PBMCs from individuals with LTBI showed decreased NF-κB phosphorylation and increased TNF-α production after HIF-1α blockade. Together, these results suggest that under hypoxic conditions, TNF-α production and NF-κB pathway downregulation are associated with the LTBI phenotype.

## 1. Introduction

Tuberculosis (TB) is an infectious disease caused by *Mycobacterium tuberculosis* (Mtb). In 2020, approximately 10 million people developed TB, and 1.3 million TB-related deaths were recorded [1]. Most individuals infected with TB are asymptomatic, with the bacteria controlled in a condition called latent tuberculosis infection (LTBI). Approximately 25% of the world population is estimated to have LTBI [2]. Additionally, 5–15% of immunocompetent individuals with LTBI will progress to active TB, with an additional 5% at risk of developing active TB at some point in the infected host’s life due to a natural decline in immunity [3]. TB infection is spread by the airborne transmission of Mtb bacilli; Mtb bacteria spread to the alveoli, where they are phagocytosed by alveolar macrophages. These phagocytes recognize Mtb through pattern-recognition receptors, such as toll-like receptor 2 (TLR-2), and engulf the bacteria [4]. 

Mtb bacilli express a wide range of virulence factors that help them survive intracellularly, including the early secretory antigen target 6 (ESAT-6) protein, which inhibits phagosome maturation [5]. ESAT-6 directly interacts with and inhibits TLR-2 [6], thereby inhibiting the production of important cytokines, including interleukin-15 (IL-15), which is responsible for mediating the activation of vitamin-D-dependent antimicrobial pathways. Inhibition of IL-15 production results in decreased activation of vitamin D, which promotes the survival of Mtb in macrophages [7,8]. Furthermore, ESAT-6 upregulates the expression in leukocytes of the A20 protein, which is a negative regulator of nuclear factor kappa B (NF-κB) that inhibits NF-κ signaling by blocking signaling pathways involving various receptors, including the tumor necrosis factor 1 receptor, CD40 receptor, TLR-2, neural retina leucine zipper, and IL-1 receptor (IL-1R). This blockade also results in decreased secretion of proinflammatory cytokines, chemokines, and nitric oxide, thereby promoting the survival of intracellular Mtb [9,10,11]. A20 is highly expressed in macrophages, as well as in mice following Mtb infection [9]. 

The NF-κB family consists of five protein monomers—p65/RelA, RelB, c-Rel, NF-κB1/p50, and NF-kB2/p52—which form homodimers or heterodimers that differentially bind deoxyribonucleic acid (DNA). These dimers are regulated in two ways: the noncanonical pathway, which is primarily activated by developmental cues; and the canonical pathway, which is primarily activated by pathogens and inflammatory mediators. The p65:p50 heterodimer is the most abundant form of NF-κB that is activated via the canonical pathway through pathogen stimulation [12,13,14,15]. The canonical NF-κB pathway is activated by various signals, including proinflammatory signals, such as those induced by IL-1R, tumor necrosis factor (TNF)-receptor family, TLRs, and T- and B-cell receptors [16]. After cellular exposure to inflammatory agents, extracellular and intracellular receptors trigger signal transduction events that activate the IKK complex by recruiting various enzymes, including kinases involved in ubiquitin chain formation [13]. NF-κB signaling is responsible for the activation of several proinflammatory cytokines, such as (TNF-α), IL-6, and IL-18, which are important for infection control. Excessive NF-κB responses are mediated by different stimuli; for instance, they can be regulated by the hypoxic response mediated by hypoxia-inducible factor 1 alpha (HIF-1α). Both HIF-1α and NF-κB play key roles in inflammatory responses and mycobacterial infections [17].

Mycobacterial diseases and hypoxia are closely associated due to the hypoxic nature of tuberculous granulomas, which are poorly vascularized, hindering the supply of nutrients and oxygenation of cells [18,19]. Hypoxia stabilizes HIF, which is a heterodimer composed of α and β subunits. The HIF-1α subunit is an important regulator of leukocytes during inflammation and various bacterial infections [20,21,22]. The dual role of HIF-1α in mycobacterial infections has been demonstrated in previous studies. According to Braverman et al. [23], HIF-1α enhanced interferon-gamma (IFN-γ)-mediated activation of macrophages in mice at the onset of Mtb infection. In contrast, it has also been reported that an increase in HIF-1α expression promoted the survival of Mtb in a mouse model of chronic Mtb infection [20].

In addition to host-cell mechanisms that establish LTBI, immune response to Mtb is crucial for the control of Mtb infection; LTBI depends on the balance between an effective immune response and the ability of Mtb to evade immune cells. Thus, in the present study, we evaluated proinflammatory cytokines, including IL-15, IL-18, IL-6, and TNF-α, which are activated by the TLR-2 receptor and mediate important pathways involved in infection control, such as the vitamin-D-dependent antimicrobial pathway, in individuals with different status of TB infection; i.e., active TB, LTBI, or healthy controls. We also evaluated the expression of the NF-κB signaling pathway, as well as A20 and HIF-1α, which are associated with the NF-κB signaling pathway. Therefore, the present study aimed to assess whether HIF-1α expression and associated mechanisms were associated with the manifestation of latent or active TB infection.

## 2. Materials and Methods

### 2.1. Ethical Statements

The present study was conducted under Resolution 466/12 of the National Health Council of Brazil, Declaration of Helsinki II (2000), and approved by the Research Ethics Committee of Universidade CEUMA, São Luís, Brazil (Protocol No. 1.803.034). Written informed consent was obtained from all study participants.

### 2.2. Chemicals and Reagents

The QuantiFERON^®^—TB Gold Plus, RNeasy Mini Kit, and SYBR Green PCR kit were purchased from Qiagen^®^ (Hilden, Germany). The High-Capacity cDNA Reverse Transcription Kit and NF-κB p65 (Total/Phospho) Human InstantOne^™^ ELISA Kit were purchased from Thermo Fisher Scientific (Waltham, MA, USA). The 2-methoxyestradiol (2-ME) was purchased from Sigma-Aldrich (Oakville, ON, Canada), and the BD OptEIA Human TNF Set Kit was purchased from BD Biosciences (San Diego, CA, USA).

### 2.3. Study Population

Patients diagnosed with active pulmonary TB were recruited at Hospital Presidente Vargas, located in São Luís, Maranhão, Brazil, from March 2019 to March 2020. We used the following inclusion criteria: positive diagnosis of pulmonary TB (characteristic symptoms, suggestive radiography, positive bacilloscopy, sputum, or bronchoalveolar lavage), written consent for participation, and correct completion of a structured questionnaire.

Intrahousehold contacts of TB patients were recruited for the LTBI and healthy control groups. Participants who tested positive in the IFN-γ release assay (IGRA) or tuberculin skin tests (TST) were considered to have LTBI. Volunteers with negative results for both IGRA and TST were included in the healthy control group, and individuals at risk of active TB were excluded.

### 2.4. Tuberculin Skin Test 

For LTBI screening, 0.1 mL of tuberculin PPD RT 23 SSI (Statens Serum Institut, Copenhagen, Denmark) was intradermally administered to the left forearm of all healthy control candidates. The skin induration reading was performed 72 h after administration. The results were based on the parameters established by the World Health Organization for positive TST (skin induration greater than or equal to 5 mm) and negative TST (skin induration lower than 5 mm).

### 2.5. IFN-γ Release Assay

For LTBI screening, 1 mL of blood was drawn into each of the following collection tubes: TB antigen tube (to evaluate the IFN-γ response to highly specific TB antigens), mitogen tube (positive control), and control tube (negative control). The tubes were incubated at 37 °C for 16–24 h. The plasma concentration of IFN-γ was determined using a ELISA method. The results were based on the parameters established by the test (QuantiFERON^®^—TB Gold) [24] and were classified as positive when the response to specific TB antigens (IU/mL) was ≥0.35% and ≥25% of nil value. Results showing a nil value of >8.0 IU/mL or a mitogen value of <0.5 UI/mL were considered indeterminate. The test was performed according to the manufacturer’s recommendations.

### 2.6. Isolation of Peripheral Blood Mononuclear Cells

Venous blood (15 mL) was collected from patients with active TB, their household contacts, and healthy controls in heparinized tubes. Peripheral blood mononuclear cells (PBMCs) were isolated using Histopaque^®^-1077 density gradient centrifugation (Sigma-Aldrich) [25]. Briefly, whole blood samples were added to a conical tube with Histopaque^®^-1077 (3 mL) and centrifuged at 900× *g* for 30 min at 20 °C. Subsequently, the buffy coat containing the PBMCs was collected. PBMCs were washed twice with a phosphate buffer solution and resuspended in Roswell Park Memorial Institute (RPMI) complete medium (Thermo Fisher Scientific, Foster City, CA, USA). The cells were counted in a Neubauer chamber, and the concentration was adjusted to 1 × 10^7^ cells/mL. After washing, the cells were resuspended in RPMI medium supplemented with 10% inactivated fetal bovine serum (Sigma-Aldrich) and 10% dimethyl sulfoxide (DMSO, Sigma-Aldrich) and stored at −80 °C.

### 2.7. Gene Expression Analysis

RNA extraction was performed using the RNeasy Mini Kit (Qiagen) as per the manufacturer’s instructions. Complementary DNA (cDNA) was synthesized using the High-Capacity cDNA Reverse Transcription Kit (Thermo Fisher Scientific). Reverse transcription-polymerase chain reaction (RT-PCR) was performed using the SYBER Green PCR kit (Qiagen) to determine the expression of *HIF1A*, *A20*, *IL18*, *IL15*, *IL6*, *TNFA*, and the housekeeping gene *GAPDH*. The final reaction volume of 20 µL contained 10 µL of master mix (SYBR Green), 7 µL of RNase-free water, 0.5 µL each of the forward and reverse primers, and 2 µL of cDNA. The primers used in the present study were purchased from DNA Express Biotecnologia LTDA (São Paulo, Brazil) and are listed below:

GAPDH:Forward: 5′-ACCCACTCCTCCACCTTTGA-3′,Reverse: 5′-CTTCTACTGGTTCAGCAGCCATCT-3′; A20:Forward: 5′-CGTCCAGGTTCCAGAACACCATTC-3′, Reverse: 5′-TGCGCTGGCTCGATCTCAGTTG-3′;IL15:Forward: 5′-GGA ATGTAACAGAATCTGGATG-3’, Reverse: 5′-GTT ATGTCTAAGCAGCAGAG-3′;IL18:Forward: 5′-ATCGCTTCCTCTCGCAACAA-3′,Reverse: 5′-CTTCTACTGGTTCAGCAGCCATCT-3′;HIF1A:Forward: 5′-CATAAAGTCTGCAACATGGAAGGT-3′,Reverse: 5′-ATTTGATGGGTGAGGAATGGGTT-3′;TNFA:Forward: 5′-CACACTCAGATCATCTTCTCAA-3′,Reverse: 5′-TTGAAGAGAACCTGGGAGTAG-3′;IL6:Forward: 5′-ATTACACATGTTCTCTGGGAAG-3′,Reverse: 5′-TTTTACCTCTTGGTTGAAGATATG-3′.

RT-PCR was performed using QuantStudio 6 (Thermo Fisher Scientific) using the following PCR reaction conditions: initial denaturation at 95 °C for 5 min; 40 cycles of denaturation at 95 °C for 15 s, annealing at 60 °C for 30 s, and extension at 72 °C for 30 s. The 2-ΔΔCT method was used for gene-expression analysis [26]. For each sample, the ΔCt was calculated by subtracting the threshold cycle (Ct) values of the housekeeping gene (*GAPDH*) from those of the target genes. Then, the following formula was used to calculate ΔΔCt values: ΔCt (target gene) − ΔCt (control gene). After determining the ΔΔCt values, the formula 2-ΔΔCT was applied to obtain the relative gene expression.

### 2.8. 25-hydroxyvitamin D_3_ (25(OH)D_3_) Levels

The serum levels of 25-hydroxyvitamin D_3_ (25(OH)D_3_) were evaluated using the chemiluminescence method-based Siemens Advia Centaur Classic kit (Siemens Healthcare Diagnostics, Tarrytown, NY, USA), as per the manufacturer’s instructions [27]. The following reference values were used to assess vitamin D status: deficiency (<10 ng/mL), insufficiency (10–30 ng/mL), sufficiency (30–40 ng/mL), and toxicity (>100 ng/mL).

### 2.9. Cell Culture and Treatment Conditions

PBMCs stored at −80 °C were thawed, transferred to a 96-well plate, and incubated at 37 °C and 5% CO_2_ for 3 h. Thereafter, the cells were incubated with 33 mmol/L of either DMSO or HIF-1α blocker, 2-ME (Sigma-Aldrich) for 1 h [28,29]. Then, the following stimuli were added: 0.5 µg/mL culture filtrate protein (CFP) from bacillus Calmette–Guérin (BCG) vaccine and 50 ng/mL phorbol 12-myristate 13-acetate (PMA). After incubating for 1 h, the culture plates were centrifuged (800× *g* for 10 min) to collect the culture supernatant, which was used for the measurement of TNF-α levels.

### 2.10. ELISA for NF-κB and TNF-α Production

A Total/Phopsho NF-κB p65 (Total/Phospho) Human InstantOne^™^ ELISA Kit (Thermo Fisher Scientific), which is a sandwich ELISA-based kit, was used to measure the levels of total and phosphorylated NF-κB, as per the manufacturer’s instructions [30]. We used cell lysates of blood samples collected from individuals with LTBI (*n* = 10) and patients with active TB (*n* = 12) to measure the levels of total and phosphorylated NF-κB. The absorbance was read using an ELISA reader (Multiskan RC/MS/EX Microplate Reader, Thermo LabSystems, Thermo Fisher Scientific) at a wavelength of 450 nm.

To determine the levels of TNF-α, we used culture supernatants collected after blockade and stimulation with CFP + PMA. TNF-α levels were determined in samples from the patients’ samples (9 LTBI and 11 TB) using the BD OptEIA Human TNF Set Kit [28], as per the manufacturer’s instructions. The optical density (OD) was measured at 450 nm using an ELISA reader (Multiskan RC/MS/EX Microplate Reader, Thermo LabSystems, Thermo Fisher Scientific). The standard curve was calculated from the readings of different concentrations of recombinant cytokines provided in commercial kits (500 pg/mL, 250 pg/mL, 125 pg/mL, 65.5 pg/mL, 31.3 pg/mL, 15.6 pg/mL, 7.8 pg/mL, and 3.9 pg/mL). The actual concentrations were calculated from the standard curve (r^2^ = 0.997) using the obtained OD values.

### 2.11. Statistical Analysis

Data are expressed as the mean ± standard deviation. ANOVA and post hoc *t*-test were used for statistical analysis, which was performed using GraphPad Prism version 6 software (San Diego, CA, USA). Differences were considered significant at *p* < 0.05.

## 3. Results

The demographic and clinical characteristics of the study participants are shown in Table 1. A total of 78 participants were recruited for this study and were distributed into three groups: 42 patients with active TB (mean age of 35.1 ± 13.8 years), 17 individuals with LTBI (mean age of 33.0 ± 13.6 years), and 19 healthy controls (mean age of 31.6 ± 10.7 years).

### 3.1. Cytokine Gene Expression Levels in PBMCs and Levels of 25(OH)D_3_

To evaluate the involvement of inflammatory cytokines and 25(OH)D_3_ in LTBI, gene expression of inflammatory cytokines and serum 25(OH)D_3_ levels were analyzed using RT-PCR and chemiluminescence, respectively. As shown in Figure 1a, the expression of *IL-15* was significantly lower in patients with TB (0.70 ± 0.06) than in individuals with LTBI (1.3 ± 0.13; *p* < 0.0001) and healthy controls (*p* = 0.0016). However, the expression of *IL-18* was significantly lower in the LTBI (0.49 ± 0.08; *p* = 0.0202) and active TB (0.65 ± 0.08; *p* = 0.0021) groups than in the control group (Figure 1b). As shown in Figure 1c, serum levels of 25(OH)D_3_ were significantly higher in patients with active TB (43.00 ± 1.7) than in individuals with LTBI (33.00 ± 1.8; *p* = 0.0052) and healthy controls (*p* = 0.0006). We also analyzed the expression of other important cytokines involved in the regulation of the immune response against Mtb infection. The expression of TNF-α was significantly lower in individuals with LTBI than in the healthy controls (−0.55 ± 0.18; *p* = 0.0104) (Figure 1d). Similarly, the expression of *IL-6* was significantly lower in individuals with LTBI than in the healthy controls (0.42 ± 0.17; *p* = 0.0371) and patients with active TB (0.66 ± 0.29; *p* = 0.0397) (Figure 1e).

### 3.2. Gene Expression of A20 and HIF1A in PBMCs

We analyzed the relationship between gene expression of *A20* and *HIF1A* in PBMCs and Mtb infection. We evaluated the level of gene expression of *A20* in PBMCs of healthy controls, individuals with LTBI, and patients with active TB. As shown in Figure 2a, *A20* mRNA expression was significantly higher in patients with TB (2.11 ± 0.27) than in healthy controls (1.11 ± 0.11; *p* = 0.0083) and individuals with LTBI (1.08 ± 0.10; *p* = 0.02).

Figure 2b shows the gene expression of *HIF1A* in PBMCs of healthy controls, individuals with LTBI, and patients with active TB. The expression of *HIF1A* was significantly higher in patients with TB than in healthy controls (*p* = 0.01). In contrast, *HIF1A* mRNA expression was significantly lower in individuals with LTBI (0.62 ± 0.11) than in patients with TB (2.66 ± 0.26; *p* < 0.0001) and healthy controls (1.57 ± 0. 28; *p* = 0.01).

### 3.3. Activation of the NF-κB Pathway in the Absence of HIF-1α

To analyze the role of the NF-κB pathway in LTBI, we evaluated the levels of total and phosphorylated NF-κB in cell lysates of PBMCs isolated from individuals with LTBI and patients with active TB before and after HIF-1α blockade with 2-ME, an estradiol metabolite that inhibits the nuclear accumulation and transcriptional activity of HIF-1α [15]. Additionally, the PBMCs were also stimulated with CFP + PMA.

No difference was observed in the expression of total NF-κB in cell lysates of PBMCs isolated from patients with active TB, individuals with LTBI, and healthy controls, regardless of stimulation with CFP + PMA or blockade with 2-ME (Figure 3a,b). However, expression of phosphorylated NF-κB was statistically different between cell lysates of PBMCs from individuals with LTBI cultured without (medium alone) or with (medium + 2-ME) HIF-1α inhibitor (*p* = 0.0081); HIF-1α blockade resulted in decreased phosphorylation of NF-κB. Additionally, there was a statistical difference in the expression of phosphorylated NF-κB between cell lysates of PBMCs from individuals with LTBI treated with HIF-1α inhibitor and cultured without (medium + 2-ME) and with (CFP + PMA + 2-ME) CFP + PMA (*p* = 0.0225); stimulation with CFP + PMA increased phosphorylation of NF-κB despite HIF-1α blockade (Figure 3c). However, no statistical difference was observed in the levels of phosphorylated NF-κB in patients with TB (Figure 3d).

### 3.4. TNF-α Levels in the Absence of HIF-1α

To evaluate the effect of HIF-1α blockade on TNF-α production, we collected culture supernatants from PBMCs cultured under the same conditions used for the evaluation of NF-κB levels. Figure 4 shows the levels of TNF-α released by PBMCs isolated from individuals with LTBI and patients with TB, prior and subsequent to CFP + PMA stimulation and HIF-1α blockade. The following groups showed statistically significant differences: The LTBI group showed statistically significant differences under the following conditions: medium + 2-ME versus medium alone (*p* = 0.0436), CFP + PMA + 2-ME versus medium alone (*p* = 0.0023), and CFP + PMA + 2-ME versus CFP + PMA (*p* = 0.0081). In the TB group, there was a statistically significant difference between CFP + PMA stimulation and medium alone (*p* = 0.0307). Furthermore, a statistically significant difference was noted between the LTBI and TB groups stimulated with CFP + PMA + 2-ME (*p* = 0.0190).

We analyzed the correlation between NF-κB expression and *HIF1A* expression in individuals with LTBI (Figure 5). Gene expression of *HIF1A* and expression of NF-κB were inversely correlated prior to HIF-1α blockade (r^2^ = 0.2951) and positively correlated following HIF-1α blockade (r^2^ = 0.6889).

## 4. Discussion

During active TB, Mtb uses virulence mechanisms to manipulate endogenous genes in infected cells (e.g., macrophages), thereby modifying essential cellular functions involved in microbicidal activity; this enhances the survival and replication of Mtb in the cells [31]. 

HIF-1α, the main regulator of hypoxic response, plays an important role in immune responses against several pathogens, such as *Staphylococcus aureus*, *Salmonella typhimurium*, *Pseudomonas aeruginosa*, and mycobacteria [21]. In addition, activation of HIF mediates various metabolic processes and the expression of antimicrobial peptides, nitric oxide, and various cytokines, such as TNFα, IL-6, and IL-12 [19,32]. In the present study, we evaluated how HIF-1α regulated the immune response and activates cellular pathways in PBMCs of patients with active TB, individuals with LTBI, and healthy controls.

IL-15 is a cytokine produced by several immune cells, such as macrophages, monocytes, and dendritic cells, and has immunoregulatory properties [33]. We observed that *IL15* expression decreased in patients with active TB, but no significant variation was observed in individuals with LTBI compared with healthy controls. The low levels of *IL-15* in patients with active TB, but not in individuals with LTBI, observed in the present study were consistent with the findings of Chandrashekara et al. [34], suggesting that Mtb infection decreased cytokine levels to survive in host cells [35].

In the present study, the observed decrease in IL-15 expression in the TB group may be attributed to the expression of ESAT-6, which directly inhibits the TLR-2 receptor responsible for the activation of IL-15. Furthermore, impairment in IL-15 expression can hinder the formation of 1,25(OH)2D_3_, as IL-15 mediates the conversion of the inactive form of vitamin D (25(OH) 2D_3_) to the active form (1,25(OH)2D_3_). This conversion further activates the vitamin D receptor (VDR), which induces the production of cathelicidin [7]. Herein, we observed a significant increase in levels of 25(OH)2D_3_ in patients with TB, which was previously reported by Owolabi et al. [27]. VDR induces the expression of antimicrobial peptides, including cathelicidin, autophagy-inducing peptides, and peptides involved in phagolysosomal fusion. This mechanism occurs through the activation of IL-15 via TLR-2 in monocytes, which in turn mediates the induction of *CYP27b1* [7]. 

Additionally, we analyzed the expression of *IL-18* in patients with TB, individuals with LTBI, and healthy controls, and found decreased levels of *IL-18* expression in individuals with LTBI. It was previously reported that the association of IL-18 with IL-12 can result in the production of cathelicidin, which is an antimicrobial peptide that activates the vitamin D-dependent antimicrobial pathway [36]. Moreover, other studies have shown that 1,25(OH)2D_3_ can modulate the functions of IL-18 and suppress its production [37]. Low levels of IL-18 are suggestive of a decreased adaptive immune response, as IL-18 plays an important role in the induction of IFN-γ. IFN-γ is associated with the T-helper type 1 profile, which is involved in the immune response against Mtb; thus, a decrease in IL-18 supports the proliferation of Mtb [38,39,40,41,42].

We also evaluated other cytokines involved in the control of active and latent TB infections. TNF-α enhances the inflammatory response, resulting in the secretion of other cytokines, including IL-1 and IL-6, and migration of immune cells to sites of inflammation; this promotes the production of chemokines and adhesion molecules [43]. TNF-α expression is important for the maintenance of tuberculous granulomas and protection against reactivation of latent TB [44]. In the present study, we observed decreased *TNFA* expression in individuals with LTBI compared with that in healthy controls. In contrast, patients with active TB exhibited increased *TNFA* expression; however, the expression was lower than that in healthy controls. The increased expression of TNF-α in PBMCs of patients with active TB compared with that in healthy individuals was also reported by Buha et al. [39]. These results highlight the importance of TNF-α in the defense against mycobacteria. 

IL-6 has proinflammatory roles that may contribute to the pathogenesis of TB. In the present study, the IL-6 expression level was lower in individuals with LTBI compared to that in healthy controls and patients with active TB. Additionally, IL-6 expression was higher in patients with TB than in healthy controls. High expression of cytokine IL-6 is a hallmark of inflammatory diseases, including TB [39,45].

The regulation of infection through proteins and transcription factors is another mechanism involved in the activation of macrophages during Mtb infection. A20 is an important immunoregulator that promotes Mtb survival in mouse infection models of TB [9]. Previous studies have reported a relationship between A20 protein levels and hypoxic conditions [46]. Hypoxia enhances Mtb survival and contributes to the LTBI phenotype owing to low oxygenation in tuberculous granulomas as the pathogen adapts to the environment [47,48,49]. Therefore, we analyzed the relationship between A20 protein and the transcription factor HIF-1α expression and Mtb infection. We observed increased *A20* expression in PBMCs of patients with active TB compared with those of healthy controls and individuals with LTBI. Previous studies reported that increased expression of A20 inhibited the activation of the NF-κB signaling pathway. Kumar et al. [9] used Mtb-infected mice and observed that let-7f miRNA-mediated regulation of A20 played a significant role in controlling the immune response of Mtb-infected macrophages. Therefore, the difference in A20 expression between individuals with LTBI and patients with active TB suggests that individuals with increased A20 expression have increased survival of Mtb bacteria.

During Mtb infection, ESAT-6 downregulates the expression of let-7f, which increases the expression of *A20*. Consequentially, this decreases the activation of NF-κB and facilitates bacterial survival through the suppression of numerous mechanisms, including apoptosis, and the production of chemokines, cytokines, and reactive nitrogen and oxygen species [9]. The observed increase in the expression of A20 in PBMCs of patients with active TB, as well as maintenance of its levels in individuals with LTBI, suggests the importance of immune-response regulators in the outcome of Mtb infection. In this context, the hypoxic microenvironment enhances Mtb survival and contributes to the LTBI phenotype due to low oxygenation in the granulomas; under hypoxic conditions, a dormant state is induced, and Mtb becomes less susceptible to antimycobacterial treatment [47,48].

HIF-1α acts in response to systemic oxygenation by regulating the expression of more than 60 genes and is also implicated in macrophage function. HIF-1α contributes to the expression of genes associated with M1 macrophage polarization, mediating the transition from a proinflammatory to an immunosuppressive phenotype while preserving the antimicrobial and protective functions of macrophages [50,51]. Our results demonstrated that the expression of *HIF1A* was decreased in individuals with LTBI and increased in patients with active TB. Consistent with our findings, previous studies have also demonstrated increased levels of HIF-1α mRNA in patients with active TB [52].

Human TB lesions are severely hypoxic, and hypoxia upregulates the expression of matrix metalloproteinase-1, a collagenase central to cavity formation and infection spread [48]. Therefore, hypoxia can contribute to the worsening of infection depending on the degree of infection. In contrast, other researchers, including Braverman et al. [23], have reported that HIF-1α expression was positively associated with IFN-γ expression at the onset of Mtb infection, which could improve the activation of macrophages. These findings suggested that HIF-1α may have a dual role in the progression of Mtb infection.

HIF-1α is associated with the NF-κB inflammatory signaling pathway [53]. In the present study, patients with active TB had higher expression of *HIF1A* than individuals with LTBI; therefore, considering the existing link between HIF-1α and NF-κB expression, we questioned the mechanism through which HIF-1α could regulate the NF-κB signaling pathway in active TB. For the same, we used 2-ME, a metabolite of 17-β estradiol that inhibits the nuclear accumulation and transcriptional activity of HIF-1α, as a tool to explore HIF-1α-mediated activation of the NF-κB signaling pathway in PBMCs isolated from patients with active TB and individuals with LTBI [20]. We observed that HIF-1α blockade decreased NF-κB phosphorylation in individuals with LTBI, without CFP + PMA stimulation. Additionally, decreased NF-κB phosphorylation was also observed in both the LTBI and active TB groups after HIF-1α blockade and stimulation with CFP + PMA. However, stimulation with CFP + PMA increased NF-κB phosphorylation in individuals with LTBI, suggesting that the presence of mycobacterial components increased NF-κB phosphorylation. No significant decrease in phosphorylation was observed in patients with TB, which may be attributed to the increase in *HIF1A* expression.

Moreover, we observed a positive correlation between *HIF1A* gene expression and NF-κB levels after HIF-1α blockade with 2-ME. These results suggested that the activation of NF-κB is not only related to HIF-1α, but other nuclear factors may also be associated with this activation during latent infection, favoring the LTBI phenotype. Bandarra et al. [17] reported a possible mechanism by which NF-κB could be activated in the absence of HIF-α, through the increased production of intracellular reactive oxygen species.

The NF-κB pathway can be activated by TNF-α-family receptors, and TNF-α is an important cytokine in the regulation of Mtb infection. Therefore, we analyzed the levels of TNF-α after HIF-1α blockade, and observed that the release of TNF-α increased in PBMCs from individuals with LTBI after blockade. However, no significant difference in TNF-α levels was observed in patients with active TB. These results suggested that both the activation of the NF-κB pathway and TNF-α production are associated with LTBI and not active TB. Hypoxia can modulate the effector functions of macrophages, which are key cells involved in the immune response against TB infection. Under hypoxic conditions, mononuclear cells and macrophages secrete large amounts of proinflammatory cytokines, such as IL-1β and TNF-α. In contrast, the absence of HIF-1α reduces inflammatory responses, phagocytic capacity of macrophages, and bacterial death [54,55].

HIF-1α plays an important role in the regulation of the NF-κB pathway, which is important in preventing excessive and harmful proinflammatory responses during infection and inflammation [17]. According to the same study, Bandarra et al., reported that HIF-1α acted by restricting the transcriptional activity of NF-κB in mammalian cells and *Drosophila*. Additionally, HIF-1α-mediated repression of NF-κB was observed when the mRNA and protein levels of A20, Cyld, DDX3, IAP1, and XIAP were measured; however, an NF-κB response was noted even in the absence of HIF-1α. NF-κB negative-feedback points showed no significant decrease [17].

The present study had some limitations; hence, further studies are needed to better understand the functions of HIF-1α in latent and active TB. For instance, the levels of the active form of Vitamin D (1,25(OH)2D_3_) were not evaluated in the present study. Differences in the levels of 1,25(OH)2D_3_ between patients with active TB and individuals with LTBI could perhaps explain the enhanced expression of IL-15 observed in individuals with LTBI; and the impossibility of carrying out the culture PBMC under hypoxic conditions using a hypoxia chamber in this study did not allow for conclusions regarding the main environments in which HIF-1α was originally expressed. Moreover, future work should address the functions of the *A20* gene in PBMCs in patients with active TB and individuals with LTBI.

## 5. Conclusions

In summary, we observed that individuals with LTBI had lower expression of *A20* and *HIF1A* and diminished levels of Vitamin D and IL-6 compared to patients with active TB. Further analysis of PBMCs revealed that blockade of HIF-1α resulted in decreased phosphorylation of NF-κB in individuals with LTBI subjects compared to the levels of phosphorylated NF-κB in patients with active TB; however, this response did not result in a decrease in TNF-α levels. These data suggested that individuals with LTBI had a higher production of TNF-α due to bypassing of the NF-κB and HIF-1α pathways (Figure 6). Furthermore, HIF-1α inhibition could be a therapeutic alternative for the treatment of tuberculosis, as it increased the immune response against Mtb.

## Figures and Tables

**Figure 1 biomedicines-10-00817-f001:**
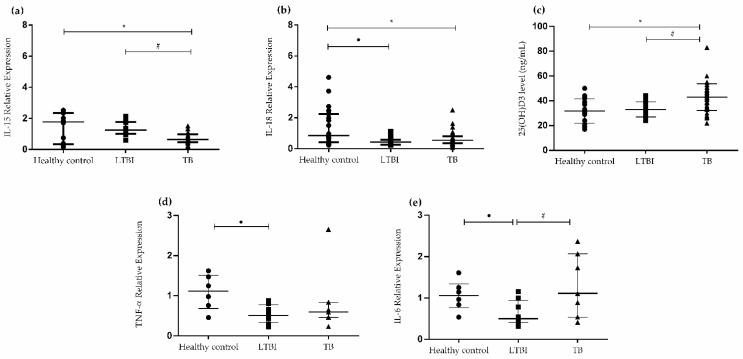
Cytokine gene expression in peripheral blood mononuclear cells (PBMCs) and serum 25(OH)D_3_ levels in patients with active tuberculosis (TB), individuals with latent tuberculosis infection (LTBI), and healthy controls. (**a**) *IL-15* mRNA expression levels in PBMCs of patients with TB, individuals with LTBI, and healthy controls. * *p* = 0.0016 for healthy control versus TB; ^#^ *p* < 0.0001 for LTBI versus TB. (**b**) *IL-18* mRNA expression levels in PBMCs of patients with TB, individuals with LTBI, and healthy controls. ^●^ *p* = 0.0202 for healthy control versus LTBI; * *p* = 0.0021 for TB versus healthy control. (**c**) Vitamin D levels were assessed for patients with TB, individuals with LTBI, and healthy controls. * *p* = 0.0006 for TB versus healthy control; ^#^ *p* = 0.0052 for TB versus LTBI. (**d**) Expression levels of *TNF-α* mRNA in PBMCs of patients with TB, individuals with LTBI, and healthy controls. ^●^ *p* = 0.0104 for LTBI versus healthy control. (**e**) Expression levels of *IL-6* mRNA in PBMCs of patients with TB, individuals with LTBI, and healthy controls. ^●^ *p* = 0.0371 for LTBI versus healthy control; ^#^ *p* = 0.0397 for TB versus LTBI. Data are presented as the mean ± standard deviation (SD) and represent one of three independent experiments. No statistically significant difference was observed between LTBI versus TB for IL-18 and TNF-α; and healthy control versus LTBI for IL-15 and vitamin D. Statistical analysis was performed using analysis of variance (ANOVA) and post hoc *t*-test, with a significance of *p* < 0.05. Abbreviations: LTBI, latent tuberculosis infection; TB, tuberculosis.

**Figure 2 biomedicines-10-00817-f002:**
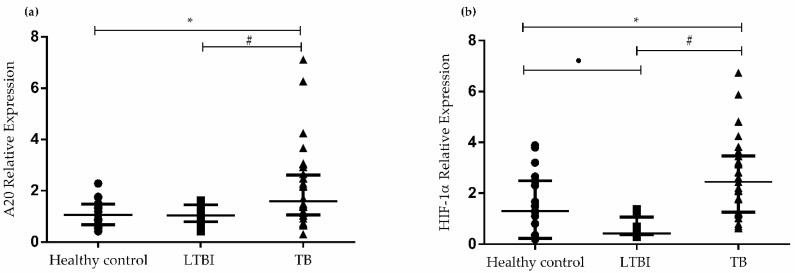
Gene expression of *A20* and *HIF1A* in PBMCs of healthy controls, individuals with LTBI, and patients with active TB. (**a**) Reverse transcription-polymerase chain reaction (RT-PCR) was performed to analyze the gene expression of *A20* in patients with TB, individuals with LTBI, and healthy controls. * *p* = 0.0083 for healthy control versus TB; ^#^ *p* = 0.0271 for TB versus LTBI. (**b**) RT-PCR was performed to quantify the gene expression of *HIF1A* patients with TB, individuals with LTBI, and healthy controls. ^#^ *p* < 0.0001 for TB versus LTBI; * *p* = 0.0135 for TB versus healthy control; ^●^ *p* = 0.0171 for LTBI versus healthy control. Data are presented as the mean ± (SD) and represent one of three independent experiments. Statistical analysis was performed using ANOVA and post hoc *t*-test, with a significance of *p* < 0.05. Abbreviations: LTBI, latent tuberculosis infection; TB, tuberculosis.

**Figure 3 biomedicines-10-00817-f003:**
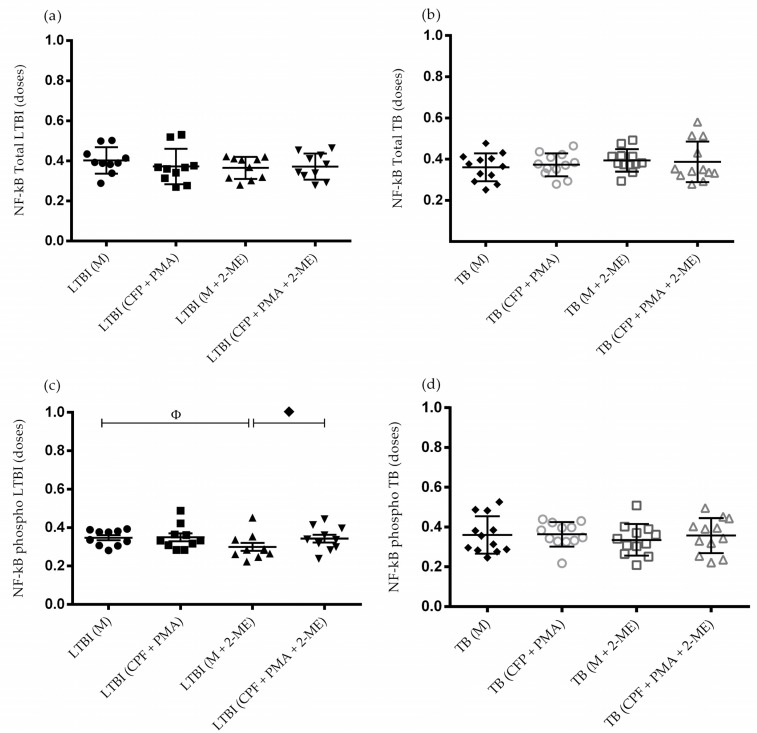
Total and phosphorylated nuclear factor kappa B (NF-κB) levels of healthy controls, individuals with LTBI, and patients with active TB. Total and phosphorylated NF-κB levels in lysates of PBMCs were evaluated using a sandwich ELISA-based kit after stimulation with culture filtrate protein (CFP) + (phorbol 12-myristate 13-acetate) PMA, under normal conditions and HIF-1α blockade with 2-methoxyestradiol (2-ME). Levels of total NF-κB in (**a**) individuals with LTBI and (**b**) patients with TB; there was no statistical difference between the groups after HIF-1α blockade. (**c**) Levels of phosphorylated NF-κB in cell lysates of PBMCs from individuals with LTBI cultured under different conditions. ^Φ^
*p* = 0.0469 for medium + 2-ME versus medium alone; ^♦^
*p* = 0.0371 for CFP + PMA + 2-ME versus medium + 2-ME. (**d**) Levels of phosphorylated NF-κB in cell lysates of PBMCs from patients with TB cultured under different conditions; there was no statistical difference between the groups after HIF-1α blockade. Data are presented as the mean ± (SD) and represent one of three independent experiments. Statistical analysis was performed using ANOVA and post hoc *t*-test, with a significance of *p* < 0.05. Abbreviations: pNF-κB, phosphorylated NF-κB; M, medium; CFP, culture filtrate protein; PMA = phorbol 12-myristate 13-acetate; 2-ME, 2-methoxyestradiol.

**Figure 4 biomedicines-10-00817-f004:**
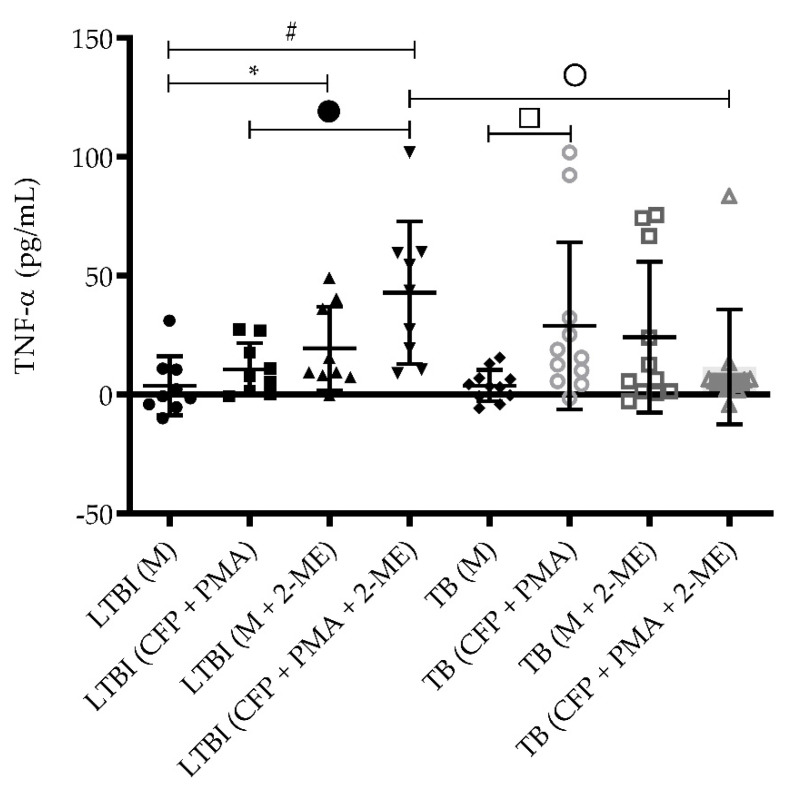
TNF-α levels in culture supernatants of PBMCs isolated from healthy controls, individuals with LTBI, and patients with active TB. TNF-α levels in PBMC culture supernatants was determined using sandwich ELISA to determine cytokine levels in individuals with LTBI and patients with TB. PBMCs were treated with 2-ME for HIF-1α blockade and stimulated with CFP + PMA. * *p* = 0.0436 for LTBI (medium + 2-ME) versus LTBI (medium alone); ^#^ *p* = 0.0023 for LTBI (CFP + PMA + 2-ME) versus LTBI (medium alone); ^●^
*p* = 0.0081 for LTBI (CFP + PMA + 2-ME) versus LTBI (CFP + PMA); ^□^ *p* = 0.0307 for TB (CFP + PMA) versus TB (medium alone); ^◦^ *p* = 0.0190 for TB (CFP + PMA + 2-ME) versus LTBI (CFP + PMA + 2-ME). Data are presented as the mean ± (SD) and represent one of three independent experiments. Statistical analysis was performed using ANOVA and post hoc t-test, with a significance of *p* < 0.05. Abbreviations: pNF-κB, phosphorylated NF-κB; M, medium; CFP, culture filtrate protein; PMA, phorbol 12-myristate 13-acetate; 2-ME, 2-methoxyestradiol.

**Figure 5 biomedicines-10-00817-f005:**
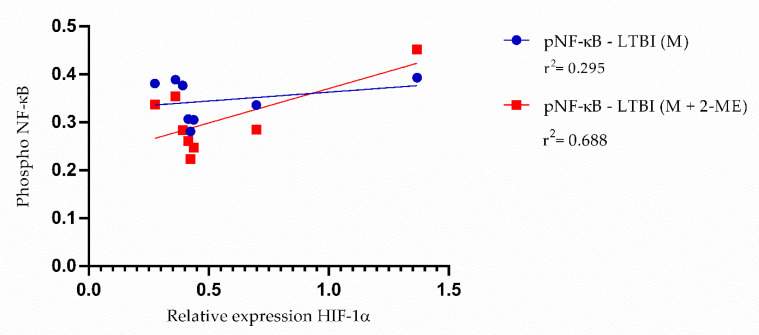
Correlation between expression of phosphorylated NF-κB and gene expression of *HIF1A* in individuals with LTBI. Significant correlations between gene expression of *HIF1A* and expression of phosphorylated NF-κB were represented by Pearson’s correlation: r^2^ = 0.2951 for pNF-κB–LTBI (medium alone), with 95% confidence interval (−0.5171 to 0.8276); and r^2^ = 0.6889 for pNF-κB–LTBI (medium + 2-ME), with 95% confidence interval (−0.03070 to 0.9381). Abbreviations: pNF-κB, phosphorylated NF-κB; LTBI, latent tuberculosis infection; M, medium; 2-ME, 2-methoxyestradiol.

**Figure 6 biomedicines-10-00817-f006:**
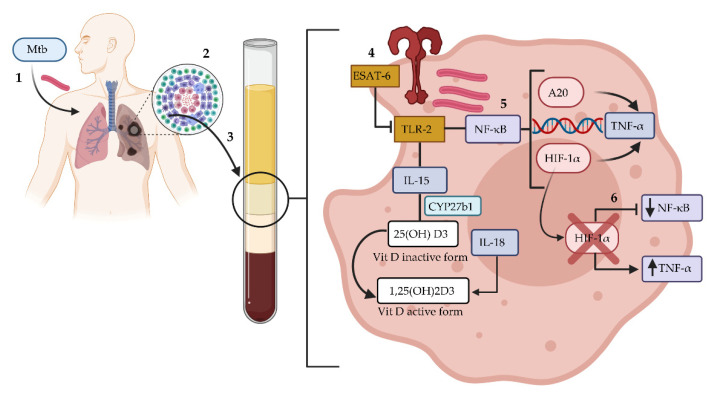
Possible mechanisms activated in PBMCs isolated from individuals with LTBI. (1) After the initial infection, the bacilli reach the lungs and are recognized by alveolar macrophages. (2) In most infected individuals, the bacilli are contained in structures called granulomas. (3) PBMCs are recruited to the site of infection. (4) *Mycobacterium tuberculosis* produces several virulence factors for cell survival, including ESAT-6, which interacts with toll-like receptor 2, activates proinflammatory cytokines, and modulates IL-15-mediated activation of vitamin-D-dependent antimicrobial pathways. (5) The NF-κB pathway activates transcription factors, such as A20, which is involved in negative feedback inhibition of NF-κB, and HIF-1α, the central regulator of the hypoxic response. (6) Inhibition of HIF-1α results in decreased phosphorylation of NF-κB but increased production of TNF-α, bypassing NF-κB and HIF-1α pathways. The figure was created using Biorender.com. Abbreviations: ESAT-6, early secretory antigen target; HIF-1α, hypoxia-inducible factor 1 alpha; IL, interleukin; Mtb, *Mycobacterium tuberculosis;* NF-κB, nuclear factor kappa B; TLR-2, toll-like receptor 2; TNF-α, tumor necrosis factor-alpha; Vit, vitamin.

**Table 1 biomedicines-10-00817-t001:** Clinical profile of healthy control, latent tuberculosis infection (LTBI), and active tuberculosis (TB) groups recruited at Hospital Presidente Vargas (São Luís, Brazil) from 2019 to 2020.

	Healthy Control	LTBI	TB
	*n* = 19	*n* = 17	*n* = 42
Gender (M/F)	6/13	4/13	30/12
Age (mean ± SD)	31.6 ± 10.7	33.0 ± 13.6	35.1 ± 13.8
TST+ (>5 mm)	7	12	–
IGRAs+	0	17	–
BAAR+	–	–	42
Culture+	–	–	12
Xpert TB/RIF+	–	–	28
X-ray+	–	–	39

–, unrealized. Abbreviations: BAAR, acid-alcohol-resistant bacillus; F, female; IGRAs, interferon-gamma release assays; LTBI, latent tuberculosis infection; M, male; SD, standard deviation; TB, tuberculosis; TST, tuberculin skin test.

## Data Availability

Not applicable.

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
