# Peer review of "Restricted Activation of the NF-κB Pathway in Individuals with Latent Tuberculosis Infection after HIF-1α Blockade"

_biomedicines, 2022, doi:10.3390/biomedicines10040817_

Round 1

Reviewer 1 Report

Restricted activation of NF-κB pathway in individuals with latent tuberculosis infection in the absence of HIF-1α

In order to bring a better version of the manuscript, authors must follow up on the recommended points and justify the evaluation. There are several concerning points that requires major revision

  1. Result format is not convenient. Result should start with what purpose or in order to investigate with the experimented result or data obtained and how much significant. All later clarification relating the results obtained has to be follow up in discussion and not in result section. Rewrite all the result section
  2. There is no conclusion given in the manuscript?
  3. Figure legend should have details of mean ± SD of three replicates independent experiments. p-value with significant comparison between groups
  4. Same significant symbols such * cannot be used for multiple group comparison, such for comparison with control if * is used then comparison with LTBI any other symbol should be in use, revise all figures
  5. All figure labels should be of same font in text.
  6. Result titles are not convenient. For example, 3.1 title should be Pro-inflammatory cytokine and Vit-D response in active TB, latent TBI patients. Figure legend 1 should have legend title.
  7. Fig 1 label should be relative mRNA expression. There should be clear explanation of IL-15, IL-18, IL-6 and Vit D expression comparison group and why there is no significant comparison between LTBI and TB for IL-18, TNF-α; control and LTBI of IL-15 Vit D; and Control with TB for IL-6? Line 23, error- in Figure mentioning

  1. 2. PBMCs of LTBI Individuals Express Less A20 and HIF-1α than those of Active TB Patients – not at all good title – change the title with aim of context

Result 3.2 line 9-14 add in introduction and only justification for discussion part

Results section in always incomplete without the justifying the obtained result that what it is represent and can suggest further. Fig 2 B. Similarly Fig 2 A not having comparison between control and LTBI?

  1. TNFAIP3 (no full name)! any protein, gene, antibody, antigen, name of test, chemical when written first in the text, there should be the full name
  2. 3. Activation of the NF-κB Pathway in Latent Infection May Depend on the HIF-1α Pathway but does not Influence TNF-α Production – rewrite the title

Fig 3, does not have negative control of Media, healthy control in the result?

Fig 3 labels should be as NF-κB (doses) & not necessary wavelength in graph. Fig 3C revise comparison group. Dose response graphs should have R2 value

  1. Fig 4 should be in separate result section. Fig 3 & 4 - comparing multiple independent variables in test should be done with ANOVA analysis
  2. Fig 5- it is not clear how the relative expression of p- NF-κB and HIF-1α expression represents in dose response relation? Significant analysis has not been mentioned nor the figure legend is written!
  3. Fig 6 should be in discussion and conclusion not in result, Fig 6 A should have specific figure mark within , it is poor to represent in the present figure on TB, LTBI and explanation and correlation for fig 6B is completely poor
  4. A discussion will justify, why the experiments have been done and many case studies so far reported and how the experimental results are suggesting, verifying and validating. There should specific paragraph explaining each of the result section obtained and not many small paragraphs.
  5. Separate section for Chemical and reagents in material & methods. Chemical in material must include company name, address, reference number in case of antibody, experimental Kit
  6. 1 title should be clinical study on tuberculosis infection with subsection TST and INF. Table 1 should be added in Material and method 2.1-2.3 section and BAAR, culture, Xpert TB/RIF, Xrays are explained in the material methods and table full form.
  7. Introduction, material method, discussion and conclusion require detailed revision. Organised meaningful writing of abstract needed
  8. There are syntax, lexical error, grammatical, spacing and symbolic errors, which requires thorough revision. Abbreviation list is missing. et al., is always italic
  9. Clinical population study should have an ethical certificate
  10. Reference section is very poor and less studied. Ref 1 does not have cited reference format

Reviewer 2 Report

Rezende et al tried to analyze the expression of cytokines and specific transcription factors that mediate mechanisms of the immune response in active TB, LTBI, and healthy control subjects. Moreover, author investigated the activation of the vitamin D-dependent antimicrobial pathway by pro-inflammatory cytokines. Author concluded that decreased production of cytokines helps to maintain latent infection and influences the activation of the NF-κB signaling pathway. The results of this study demonstrated that LTBI is multifactorial, and several factors contribute to its establishment in the host. The present study added information in this field, however, I have a few concerns regarding this study as below-

  1. In the introduction section author should discuss briefly about NFKB pathway and link with inflammation. Please add few references regarding NFKB and its role in inflammation in general.
  2. I wonder whether author performed any experiment regarding IL-1B because its active form is generated in a similar manner like IL-18? Please discuss in results on P6.
  3. Did author measured IFN-Y/IL-10 ratio which positively correlated with TST. Please see the below reference

Burl S, Adetifa UJ, Cox M, Touray E, Whittle H, McShane H, et al. The tuberculin skin test (TST) is affected by recent BCG vaccination but not by exposure to non-tuberculosis mycobacteria (NTM) during early life. PLoS ONE. (2010) 5:e12287. doi: 10.1371/journal.pone.0012287

  1. Author should add relevant references in material and methods and wherever necessary.
  2. Please proofread the manuscript carefully.

Reviewer 3 Report

The manuscript investigated and compared the levels of various pro-inflammatory cytokines in serum and PBMC cultured medium between normal, LTBI and TB patient. The conclusion and discussion is supported by the results. There are the following concerns

  1. There was less activation of these cytokines in LTBI, which did not directly influence the production of vitamin D [25(OH)2D].- not clear. Please describe how these cytokines can activate vitamin D. The formation of the active form of vitamin D has entirely different mechanism. Pro-inflammatory cytokines may affect expression of VDR.
  2. NF-κB Dosage and TNF-α Dosage should be level not dosages.
  3. Figure 1C: Low vitamin D levels are associated with TB patients. This study shows increased Vitamin D levels compared to control, please explain. Were these patients on vitamin D supplementation? What other medication these patients were on?
  4. we blocked HIF-1α to analyze the NF-κB signaling pathway.- no data to show this.
  5. we analyzed the production of TNF-α after the blockade of HIF-1α?
  6. Please subject the PBMCs to hypoxia stimulation to show the release of cytokines
  7. The molecular experiments to show the effects of vitamin D or the cytokines on each other as discussed in this article are lacking.
  8. These are the ELISA results of serum and PBMCs medium and no molecular in-vitro studies have been conducted. So please describe them accordingly or include the in-vitro mechanistic results to suport the discussion

Round 2

Reviewer 1 Report

2nd revision

Restricted activation of NF-κB pathway in individuals with latent tuberculosis infection in the absence of HIF-1α

In order to bring a better version of the manuscript, authors must follow up on the recommended points and justify the evaluation. There are several concerned points under major revision

  1. Figure image level is not changed yet nor the label fonts are similar with text font?? – for all images – labels should be of similar with text font, similar in case of bold in size and not different font size and with proper label correction.
  2. Author need to edit the title with better English meaning
  3. Abstract requires a serious revision. Each of the experiments done should be meaningful context to relate with the next. Follow up Lee et , - Pectolinarigenin Induced Cell Cycle Arrest, Autophagy, and Apoptosis in Gastric Cancer Cell via PI3K/AKT/mTOR Signaling Pathway, article, for better understanding
  4. 1 Ethical Statements should be included after author’s contribution and funding

2.2 Chemicals and Reagents – format of writing is not appropriate. Follow up for ex. Lee et  al., - Pectolinarigenin Induced Cell Cycle Arrest, Autophagy, and Apoptosis in Gastric Cancer Cell via PI3K/AKT/mTOR Signaling Pathway, article.

2.3 Title should be Population Study

2.5 when citing article for methodology reference, it must be in sync with the sentence meaning.. by the test (QuantiFERON® 6 -7 TB Gold) (24) and similar revise for 2.6

2.7 primers purchasing details not given

2.9 HIF-1α Blockade using 2-Methoxyestradiol – revise for clear explanation

2.10 and 2.11 must be in one title ELISA for NF-κB and TNF-α production

  1. Review 1st – comment no 7 to follow up. Author must read all the comments well and perform in order to reduce the same time consumption. Figure 1 image level is not changed yet nor the label fonts are similar with text font?? – for all images
  2. If the Table 1 is mentioning in the result part then start the result with meaning sentence as a story instead of “Seventy-eight participants were recruited for this study, and were distributed into…..”

And still Result dint revise the reason as asked in Review 1st – comment no 7. Reason behind the group comparison?

  1. 1st review comment not followed up nor explained??
  2. 3.3. Activation of the NF-κB Pathway in Latent Infection May Depend on the HIF-1α Pathway but does not Influence TNF-α Production – only the title is changed

Fig 3, does not have negative control of Media, healthy control in the result?

Fig 3 labels should be as NF-κB (doses) & not necessary wavelength in graph. Fig 3C revise comparison group. Dose response graphs should have R2 value

  1. In a manuscript only a similar group of comparison will have particular symbol for significant value. Here in figure 3, *# symbols used for different experiments indicating other groups.

Once different group different symbols should be followed up

Writing NF-κB total in vertical label not doses like fig 4 for TNF-α, why such variation? And similar correction required in Fig 4

  1. Conclusion is incomplete, it should justify the purpose of experiment and result obtained with validation discussed. Fig 6 image resolution should be more. Figure created by biorender.com mention in material method and remove it from image. Figure 6 fonts are not similar with text
  2. Review 1st comment 18. Add Abbreviation list

  1. Each of the topics discussed should be a sequential story rather than " X et al study or referred in most of the sentences in discussion
  2. 1st Review comment 9 not followed well. In introduction there many proteins mentioned without full form!
  3. Introduction edited-

The NF-κB family consists of five protein monomers named p65/RelA, RelB, cRel,  NFkB1/p50, and NF-kB2/p52, which form homodimers or heterodimers that binds DNA (full form)?? differentially and are regulated in two ways: the canonical pathway, dependent on the essential modulator of NF-κB NEMO –not explained well…

  1. There are syntax, lexical error, grammatical, spacing and symbolic errors, which requires thorough serious revision.

Check for plagiarism by grammerly and use english correction

Reviewer 3 Report

The manuscript has been revised  and most of the concerns have been addressed, however, the argument that TB patients are hypoxic and that's why the authors assumed that PBMCs isolated will have changed HIF-1aplha is not valid. In TB patients, TB lesions are in hypoxic condition. Yes, there may be low O2 concentration is severe cases but what about latent and early cases? So this can not be generalized. Further, isolation of PBMC and culture may change the cell properties. In-vitro and in-vivo conditions are entirely different. 

Next, the authors mentioned that they used 2-methoxyestradiol to inhibit HIF-1a and analyzed NF-kB signaling. 2-ME also inhibits NF-kB transcriptional activity. This means that 2-ME should not be used to check NF-kB activity while using it as HIF-1a blocking agent. 2-ME will inhibit bot. Please explain and justify.

Also, please include "Limitations of the study" section.

Round 3

Reviewer 1 Report

3rd review-

Manuscript revision has been done well however there are some minor revision to do

  1. A thorough correction for syntax, lexical error (especially abstract), spacing and symbolic errors to revise
  2. Conclusion correction- line 17- not culminate with reduction in the TNF- levels!

What does this study or investigation further indicates for future therapeutic understanding – add this in the conclusion end

  1. There are many paragraphs in discussion for similar context, it can be adjoined with relative context of result obtained
  2. All images don’t have same resolution and also in terms of label size (indents perfect now). Fig 6 resolution require revision and the text font inside the image is different
  3. As recommended earlier for measuring unit of NF-κB to add in result and image by understanding the standard calculation.
